# Isolation and Characterization of a Novel Pathogenesis-Related Protein-1 Gene (*AvPR-1*) with Induced Expression in Oat (*Avena sativa* L.) during Abiotic and Hormonal Stresses

**DOI:** 10.3390/plants11172284

**Published:** 2022-08-31

**Authors:** Khalid A. AlHudaib, Naimah Asid Alanazi, Mouna Ghorbel, Sherif Mohamed El-Ganainy, Faiçal Brini

**Affiliations:** 1Department of Arid Land Agriculture, College of Agriculture and Food Sciences, King Faisal University, P.O. Box 420, Al-Ahsa 31982, Saudi Arabia; 2Pests and Plant Diseases Unit, College of Agriculture and Food Sciences, King Faisal University, P.O. Box 420, Al-Ahsa 31982, Saudi Arabia; 3Department of Biology, College of Sciences, University of Hail, P.O. Box 2440, Ha’il City 81451, Saudi Arabia; 4Laboratory of Biotechnology and Plant Improvement, Center of Biotechnology of Sfax, University of Sfax, P.O. Box 1177, Sfax 3018, Tunisia

**Keywords:** environmental stress, *Avena sativa*, bioinformatic analysis, gene expression, pathogenesis-related proteins, phytohormones

## Abstract

Pathogenesis-related protein-1 (PR-1) plays crucial roles in regulating plant responses to biotic and abiotic stresses. This study aimed to isolate and characterize the first PR-1 (AvPR-1) gene in oat (*Avena sativa* L.). AvPR-1 presented conserved signal peptide motifs and core amino acid composition in the functional protein domains as the protein sequence of AvPR-1 presented 98.28%, 97.7%, and 95.4% identity with known PR1 proteins isolated from *Triticum aestivum* PRB1-2-like, *Triticum dicoccoides* PRB1-2-like, and *Aegilops tauschii* subsp. tauschii, respectively. Bioinformatic analysis showed that the AvPR-1 protein belongs to the CAP superfamily (PF00188). Secondary and 3D structure analyses of the AvPR-1 protein were also conducted, confirming sequence conservation of PR-1 among studied species. The AvPR-1 protein harbors a calmodulin-binding domain located in its C-terminal part as previously shown for its wheat homolog TdPR1.2. Moreover, gene expression analysis showed that AvPR-1 was induced in response to many abiotic and hormonal stresses especially in leaves after treatment for 48 h. This is the first study exhibiting the expression profiles of the AvPR-1 gene under different stresses in oat.

## 1. Introduction

Plants have developed different mechanisms to protect themselves from surrounding threats. These stimuli activate an array of defense mechanisms such as synthesis of antimicrobial molecules, hypersensitive response (HR), and pathogenesis-related (PR) proteins [1]. The PR proteins are thermostable and protease-resistant components which have a relatively low molecular weight of ~5–43 kDa. Moreover, the PR genes are expressed in all plant organs. Interestingly, they constitute about 5–10% of the total proteins in leaves [2,3]. PR-1 proteins are rapidly activated upon plant exposure to different biotic and abiotic stresses and form about 2% of soluble proteins in cells after infection [4]. PR genes are implicated in plant response to pathogen attack [5], wounding, jasmonic acid, salicylic acid [6,7], and ethylene [8], suggesting that PR proteins are involved in plant adaptation to different environmental stresses. Those proteins have antiparasitic activities (fungi, bacteria, viruses, insects, and nematodes) [9,10]. PR proteins are also involved in plant maturation, flowering, and plant/seed/floral development as well as leaf senescence [11].

PR proteins have been classified into 17 different families based on their main properties [12,13]. The PR-1 protein family contains the most studied proteins, known as antimicrobial peptides (AMPs), but their mode of action is still not well understood [12]. Depending on their isoelectric point, those proteins could be defined as acidic or alkaline. PR-1 proteins are generally secreted and accumulated in the extracellular/apoplastic space due to the presence of an N-terminal secretion peptide in their sequence, but many proteins could be also found in vacuoles [9]. Several PR-1 proteins have been isolated and characterized in monocot and dicot plant species such as tobacco (*Nicotiana tabacum*; [14], tomato (*Solanum lycopersicum*; [15]), durum wheat (*Triticum turgidum* subsp. durum; [7]), garlic (*Allium sativum* L.; [4]), and banana (Musa spp.; [16]). All known PR-1 proteins have a conserved cysteine-rich secretory protein, antigen-5, and pathogenesis-related-1 (CAP) domain. This domain typically folds into four α-helices and four β-sheets stabilized by disulfide bonds. Such unique structures are indispensable for their biological roles in response to different biotic and abiotic stresses [17,18,19]. In *Allium sativum* L., three different PR genes (*PR1*, *PR3*, and *PR5* genes) are positive marks for garlic resistance to *F. oxysporum* f. sp. *cepae* [10]. Furthermore, it has been shown that garlic infection with different pathogenic fungi of the genus *Fusarium* induced the expression of different pathogen-related protein genes such as *AsPR1* (*AsPR1c*, *d*, *g*, *k) and AsPR2 (AsPR2b*, *AsPR5a, c)* in roots and *AsPR4* (*AsPR4a(c)*, *b)* and *AsPR2c* in stems and cloves of both resistant and sensitive cultivars [4]. In sugarcane (*Saccharum* spp. hybrids), 19 different PR-1 proteins were identified that respond to a wide range of stresses such as infection with *Acidovorax avenae* subsp. *Avenae* (*Aaa*), as well as other different abiotic stresses such as NaCl, PEG6000, and SA treatments [20]. In *Arabidopsis*, it has been shown that an ELF18-INDUCED LONG NONCODING RNA 1 (ELENA1) acts as a positive regulator of immune responsive genes during their transcription [21]. Moreover, ELENA1 is associated with mediator subunit 19a (MED19a) to enhance enrichment of the complex on the *PATHOGENESIS-RELATED GENE 1* (*PR1*) promoter, whereas FIBRILLARIN 2 (FIB2) is negative transcriptional regulator of many immune responsive genes, such as *PR1*. ELENA1 can dissociate the FIB2/MED19a complex. Thus, it releases FIB2 transcriptional regulator from the *PR1* promoter and enhances *PR1* expression [21]. SlPR-1, a tomato PR-1 protein, was induced after plant treatment with SA and infection with *Meloidogyne incognita* nematode [6]. In tomato, plant infection with *Alternaria solani* induced the expression of a pathogenesis-related protein-like protein gene (known as BG124298). This gene was upregulated 5.57-fold and 1.63-fold in a resistant and a susceptible genotype, respectively [22]. *PR-1* genes also have crucial roles in response to abiotic stresses. In *Triticum aestivum*, it has been demonstrated that plant treatment with glycerol induced the expression of different genes such as pathogenesis-related (PR) genes (PR-1, PR-3, PR-10) and peroxidase [23]. In addition, *TaPR-1-1* gene expression was induced by osmotic stresses, freezing, and salinity. Interestingly, overexpression of *TaPR-1-1* positively regulated plant tolerance to those stresses in yeast and *Arabidopsis* [24]. Despite the extensive work on PR-1 proteins, little is known about their regulation. In *Arabidopsis*, it has been demonstrated that PR-1 promoter interacts with AtWRKY50 via its C-terminal part. AtWRKY50 is considered as the most effective WRKY activator of PR1 gene expression [25]. This interaction occurs simultaneously in presence of TGA2 or TGA5, and AtWRKY stimulates this binding [25]. The same result was also found in tobacco [26] and banana [27]. More recently, it has been demonstrated that durum wheat TdPR1.2 physically interacts in vitro with CaM/Ca^2+^ complex. This interaction enhances the catalytic activity of TdPR1.2, which is further enhanced in presence of Mn^2+^ cations [7]. Moreover, TdPR1.2 confers abiotic stress tolerance (salt, osmotic, and heavy metal stress) to *E. coli* [28].

*Avena sativa* plants are found all over the word. Those cereals are rich in health-promoting substances and can be used as animal feed, forage, and food [29,30]. Furthermore, oat is more resistant to salty soils compared to other cereals. *Avena sativa* is an important crop for improving plant adaptability to salty alkali soils [31]. Many studies investigated gas exchange [32], oxidative stress [33,34], and ions [32,33,34,35,36] of oat plants subjected to salt stress. Recently, it has been shown that application of exogenous Ca^2+^ can alleviate salt stress applied to oat grown in salty mediums [31] by conserving the stability and functions of the plasma membrane in cells [37]. Although an important number of gene families are implicated in the resistance and defense of plants against biotic and abiotic stresses, PR-1 genes have not been revealed in oat plants, and their biological functions remain largely unknown. In the present study, we isolated and characterized a novel PR-1 gene from oat (*Avena sativa*) and studied its tissue expression patterns in response to different abiotic and hormonal stresses. Our results provide new insights into the function of the *A. sativa* PR-1 gene, which can be used in breeding programs to increase the resistance to different abiotic stresses in cultivated *Avena* spp.

## 2. Results

### 2.1. AvPR1 Sequence Analysis

Sequence analysis of the AvPR-1 gene (GenBank OP132412) revealed an ORF length of 525 bp. Expasy tools analysis showed that the corresponding protein presented a size of 174 amino acids with a predicted molecular weight of 18.89 kDa and an isoelectrical point (pI) of 9.19 (Table 1). The aliphatic index (AI) was 63.45. This index reflects the relative number of hydrophobic residues. Finally, the GRAVY index of AvPR-1 was negative (−0.288), which means that AvPR-1 is predicted to be a hydrophobic protein [4]. All studied PR-1 proteins have a negative GRAVY index, which means that all those proteins are hydrophobic (Table 1). As shown in Table 1, different PR1 proteins isolated so far have a similar length (161–179 aa) and a predicted Mw of 17–19 kDa. The novel PR-1 protein is a basic protein as shown for many other PR-1 proteins isolated from other different species [7,15,16].

Next, analyses of the AvPR1 amino acid sequence were performed using the NCBI server. The result showed that AvPR1 belonged to the cysteine-rich secretory proteins, antigen-5 and pathogen-related protein-1 (CAP) superfamily; Figure 1a. AvPR1 protein contains a putative conserved SCP_PR-1-like domain (cd05381) of 135 aa (from aa 29 to aa 164 (pfam00188)). AvPR-1 harbors a caveolin-binding domain (CMB; 108–113; Figure 1b). CBM represents the putative sterol-binding domain identified in pathogen-related yeast 1 (PRY1) proteins [17]. Moreover, CAP-derived peptide (CAPE) with conserved residues was identified (154–160; Figure 1b) [38]. Such domains were previously identified in many PR-1 proteins from different species such as pepper, banana, soybean, and tomato [1,15,39,40].

The Signal P-5.0 database revealed the presence of a cleavage site between positions 24 and 25. In addition, signal peptide is found in the AvPR-1 structure. This site is the 8 aa Signal Peptide found in the N-terminal part of the AvPR-1 protein (positions 1–8; Figure 1b and Figure 2). Sequence analysis of the different studied proteins shown that those signatures are conserved among PR-1 proteins (Figure 2). Using the PONDR database, analysis showed a second site in the C-terminal part of AvPR-1 (173–174; Figure 2). The C-terminal locates signal peptides of basic PR-1 proteins, controlling protein transport into vacuoles [41] and is constitutively expressed by stress signals [1]. The same results were also found in TdPR1.2 [28]. The presence of the transmembrane region in the AvPR-1 sequence was also investigated using the PDONR database (https://services.healthtech.dtu.dk/service.php?TMHMM-2.0). The AvPR-1 sequence harbors a transmembrane region in the N-terminal part of the protein (6–24 aa, Figure 1b, Appendix A). Overall, our results are coherent with some previously reported findings for other species [20,39,42]. A conserved calmodulin binding domain was mapped in the AvPR-1 structure using the calmodulin target database (data not shown). This domain is located at the C-terminal part of the protein. A similar domain was recently identified in the TdPR1.2 protein that can interact with calmodulins in a calcium-dependent manner [7].

### 2.2. Phylogenetic Analysis of AvPR-1

Different PR1 protein sequences isolated from different plant species were obtained from NCBI. Sequence analysis of those proteins using Cluster Omega revealed that those proteins are highly conserved and present an important sequence homology (Figure 2). The deduced protein sequence of AvPR1 shared a high similarity with other PR-1 proteins, ranging from 98.28% identity with bread wheat (*Triticum aestivum* PRB1-2-like, GenBank accession number XP_044433901.1) and durum wheat (*Triticum turgidum* subsp. durum TdPR1.2 (GenBank accession no. MK570869.1) PR-1 proteins to 97.7% and 95.4% identity with PR1 proteins isolated from *Triticum dicoccoides* PRB1-2-like and *Aegilops tauschii* subsp. *tauschii* (GenBank: XP_020170282.1), respectively.

Moreover, phylogenetic tree analysis was performed using the same database through the neighbor-joining method. This resulted in five major clusters, viz. I, II, III, IV and V. As expected, PR1 proteins isolated from dicotyledonous plants were clustered into one group (group I, Figure 3a), whereas proteins isolated from monocotyledons were clustered into four groups. The second group is formed by proteins isolated from *Phoenix dactylifera* and *Musa acuminate*. The majority of proteins were clustered into group III, whereas AvPR1 protein was clustered with barley and *Aegilops taushii* proteins (cluster IV), suggesting an evolutionary conservation of those species and that oat protein may share a common ancestor with those proteins and could perform the same functions. In addition, the last group was formed by proteins isolated from three wheat plants (Figure 3a).

To create LOGO motifs of the AvPR-1 protein, MEME was run on the sequence database. The LOGO representations of the protein are shown in Figure 3b. Analysis showed 10 conserved domains in PR-1 protein sequences. Those motifs are conserved among studied species (Figure 3b, Appendix A). As shown in Appendix A, five domains (blue, red, green, cyan, and orange) are much conserved among studied PR-1 proteins. The SGDLSG motif (yellow) is found only in PR-1 isolated from dicotyledonous plants used in this work and absent in all monocotyledonous plants. The MNFTNYSRFLIVF motif (pink) was found only in four dicotyledonous plants. 

Protein phosphorylation is an important post-translational modification (PTM) controlling crucial cellular processes, such as signaling, transport, and nutrient uptake [43]. Thus, the number of putative phosphorylated sites was also investigated using the Notphos 3.1 server. AvPR-1 presented 12 putative phosphorylated residues (S3, S4, S7, S24, T28, T49, Y61, S86, S89, S98, S130, and T131, Appendix A). Such a result may prove that AvPR-1 is phosphorylated in cells in response to different stress conditions. Finally, the GPS-SNO predictor was also used to predict the putative nitrosylation residues in the PR-1 structure. Analysis shows that the AvPR-1 sequence harbors six different nitrosylation sites (C69, C113, C119, C135, C140, C150). Those residues are crucial for protein post-translational modifications [44].

### 2.3. Gene Ontology and KEGG Annotation

Gene ontology (GO) was carried out based on biological process, molecular function, and cellular component terms for the AvPR-1 protein using the PANNZER2 online server (Appendix A). The AvPR-1 protein regulates two biological processes (GO:0006952 defense response and GO:0009607 response to biotic stimulus) and two cellular components (GO:0005576: Extracellular region and GO:0016021: Integral component of membrane). A KEGG orthology analysis of AvPR-1 revealed that oat PR-1 gene was mapped to the MAPK signaling pathway (04016), plant hormone signal transduction (04075), and to the plant–pathogen interaction (04626). Such findings were also cited for other PR-1 proteins such as *P. nigrum* [1]. Based on the gene ontology enrichment data, the oat PR-1 protein function suggests that the *PR-1* gene has an important role in plant defense against different abiotic stress treatments.

### 2.4. Interaction Network of AvPR-1 Protein

The interaction network of the AvPR-1 protein from oat was constructed based on the interaction relationship of the homologous PR-1 proteins from *Triticum aestivum* (Figure 4). The interaction network analysis showed that the AvPR-1 protein interacted with the Glyco_18 domain-containing protein, which belongs to the glycosyl hydrolase 18 family, a Bet_v_1 domain-containing protein (uncharacterized protein) and a Pex2_Pex12 domain-containing protein (uncharacterized protein). Glycoside hydrolase family 18 (GH18) belongs to the chitinase subfamily. It catalyzes the degradation of β-1,4 glycosidic bonds in amino polysaccharides and possesses different functions. GH18 chitinases are implicated in many physiological processes, such as nutrition uptake and regulation of the immune response [45].

### 2.5. Predicted Secondary and 3D Structures of the AvPR-1 Protein

The secondary structure analyses of AvPR-1 were performed using the SOPMA online server (Figure 5a, Table 2). AvPR-1 has 55 α-helices, 8 β-turns, 81 random coils, and 30 extended strands (Table 2).

The same result was observed in TdPR1.2 and TaPR-1 proteins (Table 2, Appendix A). The predicted 3D structures of AvPR1 protein and other studied proteins were generated online using the Phyre2 server (Figure 5b). The 3D structure of the AvPR-1 protein is conserved, especially regarding the α–β–α sandwich structure that is characteristic of PR-1 proteins (Figure 5b) [42].

### 2.6. Differential Expression of AvPR-1 Gene under Various Stress Conditions

To investigate the possible biological functions of the *AvPR-1* gene, we assessed the expression patterns of AvPR-1 genes in oat under various abiotic stress conditions using qRT-PCR (Figure 6 and Figure 7).

In response to salt stress (150 mM NaCl), AvPR-1 was significantly upregulated (Figure 6a). The same result was observed when plants were subjected to mannitol and PEG stresses (Figure 6b,c). When heat stress was applied to oat plants (42 °C for 30 min), there was a significant increase in AvPR-1 expression level in roots and shoots, suggesting that this protein could have a putative protective role in controlling oat heat tolerance (Figure 6d).

The hormonal response of the AvPR-1 gene was investigated by treating plants with salycilic acid (SA), indole acetic acid (IAA), jasmonic acid (JA), and abscisic acid (ABA). As shown in Figure 7, AvPR-1 was upregulated in response to all hormones used in this work. Overall, those results demonstrate that AvPR-1 is implicated in plant response to many abiotic and hormonal stresses.

## 3. Discussion

Cereals have a key role in fulfilling the world’s food demand [24]. They are exposed to variable stresses during their life cycle [24], and this is especially the case for the common oat *Avena sativa*. Thus, a crucial role of *PR-1* genes is necessary for the pathogenesis-related metabolic pathways. PR-1 genes have various intricate growth/developmental mechanisms. They also help plants to cope with an important number of environmental stresses. *PR-1* genes are among the proteins that show a high level of transcription in response to biotic and abiotic stress applications. These proteins have a crucial role in plant defense against biotic stresses as they thicken the cell wall to block the apoplastic spread of pathogens [46]. Furthermore, PR proteins are crucial components in cells that enhance plant response to an important number of abiotic stresses such as light, salt, low temperature, and drought [9,42,47]. Thus, those genes are used to generate transgenic plants that have an enhanced tolerant to different pathogens such as oomycetes [48], bacteria [49], and fungi [50], as well as abiotic stresses.

The PR-1 gene family has been identified and characterized in many plant species. The number of PR-1 family members differs depending on the species. For example, 13 genes were identified in tomato [15], 19 in *Saccharum*
*spontaneum* [20], and 23 genes in wheat and rice [51,52]. Despite their importance in plants, little is known about the PR-1 genes in monocotyledons, especially in oat. In the current study, the first PR-1 gene was identified in oat. AvPR-1 presents a negative GRAVY score, which is common for hydrophilic proteins. The AvPR-1 protein possess the CAP signature, the CBM domain is involved in sterol binding [17], and the CAPE is involved in plant immune signaling [38]. In general, the ability of the PR-1 proteins to bind sterols is correlated with their antimicrobial activity against the sterol auxotroph, a major plant pathogen known as the Phytophthora species [53]. CAPE-1 is well conserved among the monocots and dicotyledonous plants. The consensus motif PxGNxxxxxPY is found in AvPR-1. Sequence analysis of AvPR-1 structure revealed the presence of a calmodulin-binding domain in the C-terminal part of the protein. Such a domain was also identified in durum wheat TdPR1.2 [7]. Calmodulins (CaMs) are ubiquitous Ca^2+^ bonding proteins highly conserved in eukaryotes that decode the Ca^2+^ signaling pathways in plants. After plant exposure to stress, the intracellular Ca^2+^ concentration increases. This variation in Ca^2+^ level is perceived by CaMs (and other calcium sensors) leading to the formation of active Ca^2^^+^/CaM complexes able to interact with a variety of target proteins (phosphatases, kinases, catalases, PR-1s) [7,54,55,56].

The presence of these motifs strongly suggests that AvPR-1 responds to different biotic stresses [38,40,42,57]. In this study, AvPR1 was upregulated in leaf and root tissues of a Saudi oat cultivar subjected to NaCl (150 mM) and PEG (10% PEG 6000) treatments as previously shown in other studies such as in *Zea mays* (ZmPR-1; [55,58]), banana [16], tomato (13 SlPR-1 genes; [15]), *Vitis vinifera* (VvPR-1; [59]), and rice (OsPR1a; [19]). Moreover, ScPR-1 was upregulated in leaf tissues of *Saccharum spontaneum* after salt and PEG stresses but downregulated in stem (MT11–610 cultivar) after PEG treatment and root tissues (ROC22 and MT11–610 cultivars) after NaCl treatment [20], suggesting that this gene could have a dual role depending on the tissue expression in the plant. Many studies have described the effect of heat stress on plant [60,61].

It was described in the literature that the SA and JA signaling pathways are stimulated after biotrophic/hemibiotrophic (under the control of SA) and necrotrophic (under the control of JA) pathogen infection [9]. Thus, we investigated the effect of SA and JA application on AvPR-1 gene expression in oat. Our results showed that AvPR-1 was upregulated after application of those phytohormones in roots and shoots of oat.

Other different PR-1 proteins were reported to be upregulated after plant treatment with SA and JA [15,24]. In banana, MaPR1-1 was upregulated after plant treatment with SA and JA stresses due to the presence of cis-elements and binding sites for transcription factors [27]. Thus, identification of stress-responsive elements involved in up/downregulation of PR-1 will help in understanding plants’ resistance mechanisms toward various stresses. These findings strongly suggest that the AvPR-1 gene plays a crucial role in plant defense against environmental stresses. It has been suggested that PR-1 genes can serve as molecular markers associated with resistance to different biotic and abiotic stresses [1,4,9]. Thus, our findings could be useful for breeding programs aimed at increasing the resistance of oat crops to salt, drought, and hormonal stresses as well as plant infection with pathogens. This could be achieved by, for example, generating an oat crop that overexpress the AvPR-1 gene, which may protect against surrounding environmental threats. In addition, besides PR-1 genes, plant defense against environmental stresses is associated with some other PR genes from other families such as chitinases and endo-1,3-_-glucanases, TLPs, and PR-10.

## 4. Materials and Methods

### 4.1. Plant Material and Stress Treatments

In this work, seeds of Saudi oat (*Avena sativa* L.) were kindly given from a private field in Ha’il, Saudi Arabia. Before incubation, around 45 seeds were sterilized in each box containing 30 mlof 0.6% NaClO solution for 15 min, then washed five times with 50 mL sterile water. For each treatment, 45 seeds were placed in each Petri dish (11 cm long, 2.5 cm high and 11 cm wide) in the presence of a sponge and filter paper placed below to maintain moisture at 25 ± 2 °C. Seeds were then transferred to a greenhouse at 24 ± 2 °C, with photosynthetically active radiation of 280 μmol m^−2^ s^−1^, a 16 h photoperiod, and 60 ± 10% relative humidity. After 10 days, seeds were subjected to stresses. In this study, nine treatments were used including the control (distilled water), 150 mM NaCl, 10% PEG, 200 mM mannitol, 5 mM of each phytohormone (SA, JA, IAA and ABA), and heat (42 °C). Each treatment was replicated three times. Finally, shoots were harvested and immediately frozen in liquid nitrogen and stored at −80 °C.

### 4.2. Isolation of the cDNA AvPR1.2

Total RNA was isolated from *Avena sativa* plants subjected to 5 mM SA for 24 hours using the Trizol reagent (Invitrogen) according to the manufacturer’s protocol. Next, the remaining genomic DNA was removed using RNase-free DNase. First-strand cDNA was synthesized from 2 μg of total RNA using M-MLV reverse transcriptase (Invitrogen), according to the manufacturer’s protocol.

Sequence alignment of different full sequences of PR1 genes was performed. Primers corresponding to 5′ and 3′ UTR regions were designed and used to amplify AvPR1. After cloning and sequencing of the fragment, the AvPR1 full-length cDNA was amplified using two specific primers AvPR1_Fr (5′-ATGGCA TCT TCC AAG AGC AG -3′) and AvPR1_Rv (5′-TCA AGGGTG AGG ACG CGA A-3′), designed based on *Triticum turgidum* subsp. durum; *TdPR1.2* sequence (MK570869.1). PCR products were purified from agarose gel, cloned into pGEM-T Easy vector, sequenced using an ABI PRISM automated sequencer, and then published (GenBank OP132412).

### 4.3. Sequence Analysis of AvPR1

The predicted protein was characterized using the ProtParam server (http://web.expasy.org/protparam; accessed on 1 May 2022) [62] to investigate MW, pI, AI, and GRAVY. The conserved domains, sites, and motifs were analyzed using NCBI-CDD (https://www.ncbi.nlm.nih.gov/cdd; accessed on 1 May 2022). Multiple sequence alignment and phylogenetic tree analysis were conducted with Cluster Omega (https://www.ebi.ac.uk/Tools/services/web/toolresult.ebi?jobId=clustalo-I20220527-204434-0929-25108385-p2m&analysis=phylotree). The signal peptide cleavage sites (SignalP 5.0; http://www.cbs.dtu.dk/services/SignalP/; accessed on 27 May 2022) were also used to detect the presence of putative signal peptides in the AvPR-1 protein [63].

The AvPR1 sequence-predicted phosphorylation sites of the AvPR-1 protein were identified using the NetPhos 3.1 server (https://services.healthtech.dtu.dk/service.php?NetPhos-3.1) [64]. The conserved motifs of the AvPR-1 protein were identified using the Multiple Em for Motif Elicitation (MEME) server v5.1.1 (http://meme-suite.org/tools/meme) [64]. For MEME searches, the “any number of repeats” mode was used, with a search limit of 10 motifs. All other parameters were left at their default values.

The trans-membrane helix was identified using the TMHMM database (https://services.healthtech.dtu.dk/service.php?TMHMM-2.0). Predicted disordered regions (DRs) were identified using the Predictor of Natural Disordered Regions (PONDR) server (http://www.pondr.com/). The physiological role of AvPR-1 was revealed by Pannzer2 (http://ekhidna2.biocenter.helsinki.fi/sanspanz/). The subcellular localization of the AvPR-1 protein was predicted by the WoLF PSORT II online software (https://www.genscript. com/wolf-psort.html?src=leftbar). The presence of a putative calmodulin-binding domain was revealed by the calmodulin target database (http://calcium.uhnres.utoronto.ca/ctdb/no_flash.htm).

### 4.4. Secondary and Tertiary Structure Analyses

Secondary structure analyses were performed using the SOPMA server (https://npsa-prabi.ibcp.fr/cgi-bin/npsa_automat.pl?page=/NPSA/npsa_sopma.html) [65]. Predicted 3D structure analyses were generated using the Protein Homology/analogY Recognition Engine v2 (Phyre2) server (http://www.sbg.bio.ic.ac.uk/~phyre2/html/page.cgi?id=index) [66].

### 4.5. Gene Ontology (GO) Analysis

Gene ontology (GO) was used to obtain information about the *AvPR-1* gene involvement, including biological processes, cellular components, and molecular function using the PANNZER2 web server (http://ekhidna2.biocenter.helsinki.fi/sanspanz/).

### 4.6. Interaction Network of AvPR-1 Proteins

Protein–protein interaction (PPI) was studied using STRING v11.0 for PR-1 using *Triticum aestivum* as reference [67]. The minimum required interaction score parameters were set at the medium confidence level.

### 4.7. RNA Extraction and Quantitative Real-Time Reverse Transcription PCR (qRT-PCR)

Total RNA was extracted from individual roots and leaves (0.5 g of each tissue) using the RNeasy Plant Mini Kit (QIAGEN, Hilden, Germany). Extracted RNA was then purified from genomic DNA (RNase free DNase set; QIAGEN), qualified by gel electrophoresis, and used for first-strand cDNA synthesis (GoScript Reverse Transcription System; Promega, Madison, USA) with an oligo-dT primer. PCR reactions were achieved in a 10 μL final volume tube in the presence of 3 μL cDNA (obtained from 40 ng of DNase-treated RNA), 0.5 μL of each primer of the AvPR-1 gene (AvPR_Fw and AvPR_Rv at 10 μM), 5 μL 2 × SYBR Green I master mix and 1 μL of RNase-free water (Sigma). The reaction consisted of an initial denaturation at 95 °C for 5 min followed by 40 cycles composed of 10 s at 95 °C, 20 s at 60 °C, and 30 s at 72 °C, then a melting curve (5 s at 95 °C, 1 min at 65 °C, and 5 min with the temperature increasing from 65 to 97 °C). Three biological repetitions were performed for each experimental condition, with three technical repetitions for each sample. Melting curve analysis at the end of cycling was used to verify whether there was single amplification. At the end of the reaction, the threshold cycle (CT) values of the triplicate PCRs were averaged and used for transcript quantification. The relative expression ratio of the *TdPR1.2* gene was calculated by using the comparative CT method with the *actin* gene designed from the *T. aestivum* genome (actin Av_Fw: 5′-TCC CTC AGC ACA TTC CAG CAGAT-3 and actin Av_Rv: 5′-AAC GAT TCC TGG ACC TGC CTC ATC-3′) as an internal expression standard [68]. The relative expression level was calculated from triplicate measurements based on the 2-DDCT, where DDCT = (CT, target gene−CT, actin) stressed—(CT, target gene−CT, actin) control. Relative expression ratios from three independent experiments (three biological repetitions) are reported.

### 4.8. Statistical Analysis

Data are reported as mean ± S.E. The results were compared statistically by using Student’s *t* test, and differences were considered significant at *p* < 0.01.

## 5. Conclusions

Pathogenesis-related protein-1 (PR-1) is the most produced protein during plant response toward many environmental stresses. Moreover, PR-1 genes play a crucial role in plant growth and maturation. However, the PR-1 gene family in oat has not been previously studied. This study provides a comprehensive understanding of the first isolated PR-1 gene from oat (*Avena sativa*), including gene structure, phylogenetic relationship, motifs, and gene expression profiles against different stresses. The structural analysis of AvPR-1 revealed similar binding pockets in the predicted 3D structures of other different PR-1 proteins. Expression analysis of AvPR-1 showed a positive correlation, and the identified candidate PR-1 gene must be further functionally validated for its biological significance and molecular mechanisms. It could be also a hopeful candidate for selecting multiple stress tolerant oat varieties.

## Figures and Tables

**Figure 1 plants-11-02284-f001:**
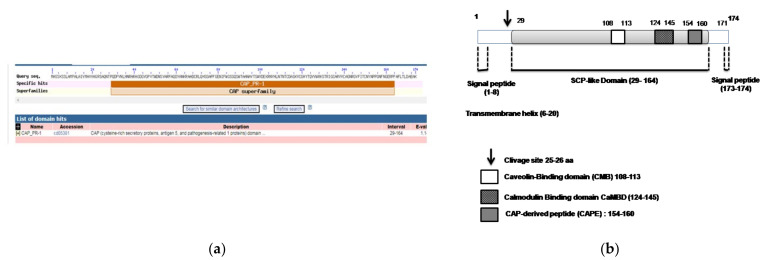
Bioinformatic analysis of AvPR-1 structure. (**a**) Analysis of AvPR-1 proteins using the NCBI server showed that AvPR-1 belongs to the CAP superfamily. (**b**) Conserved domains of the AvPR-1 protein. The predicted AvPR-1 protein contained a conserved motif at residues 29–164 aa that belonged to the CAP-superfamily. Two peptide signals were also identified in the N-(1–25) and C-(164–174) terminal parts with the presence of a transmembrane domain. AvPR-1 also contains a cleavage site at position 25.

**Figure 2 plants-11-02284-f002:**
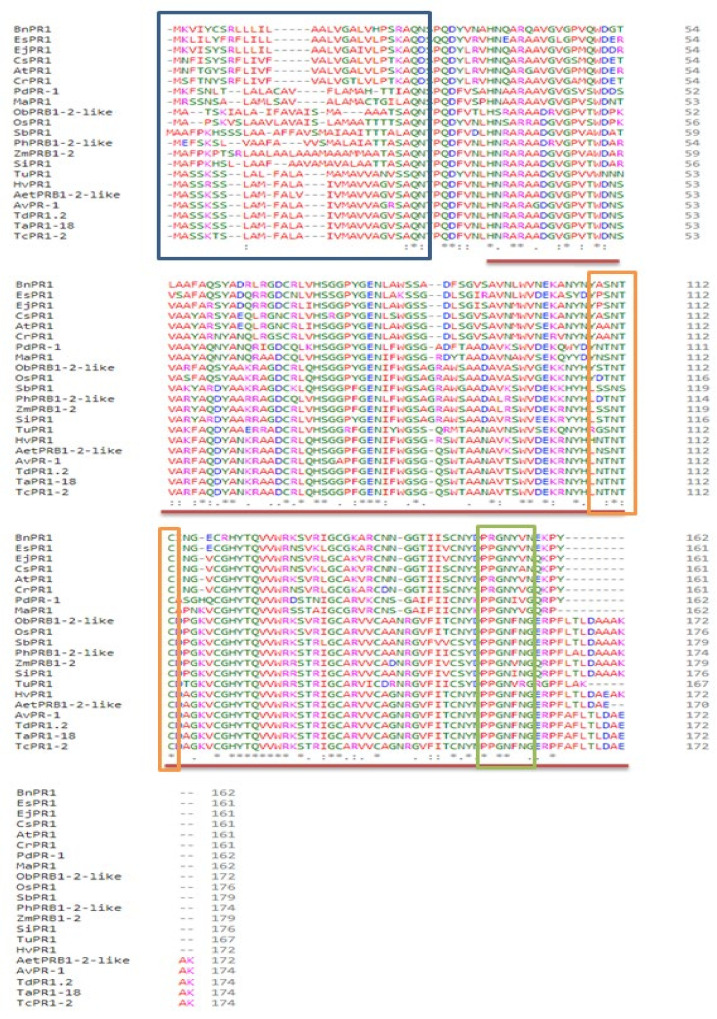
Sequence alignment of different PR-1 proteins identified in oat (GenBank OP132412), *Triticum turgidum* subsp. durum (MK570869.1), *Triticum aestivum* (XP_044433901.1), *Aegilop stauschii* subsp. tauschii (XP_020170282.1), *Triticum urartu* (EMS45472.1), *Phoenix dactylifera* (XP_008796972.2), *Hordeum vulgare* subsp. vulgare (BAK01044.1), *Triticum dicoccoides* (XP_037463373.1), *Panicum hallii* (XP_025795067.1), *Oryza brachyantha* (XP_006661674.1), *Zea mays* (XP_008657154.1), *Eutrema japonicum* (BAF03626.1), *Camelina sativa* (XP_010467245.1), *Arabidopsis thaliana* (NP_179068.1), *Eutrema salsugineum* (XP_006409652.1), *Capsella rubella* (XP_006299028.1), *Brassica napus* (XP_013733404.1), *Sorghum bicolor* (XP_002465112.1), *Setaria italica* (XP_004983398.1), *Musa acuminata* (ABK41053.2), and *Oryza sativa* Japonica Group (XP_015613013.1). The blue rectangle indicates the signal peptide, the PF00188 domain structure is highlighted with a red line, caveolin-binding motif (CBM) is highlighted in orange, and the CAP-derived peptide (CAPE) with conserved residues is highlighted in green.

**Figure 3 plants-11-02284-f003:**
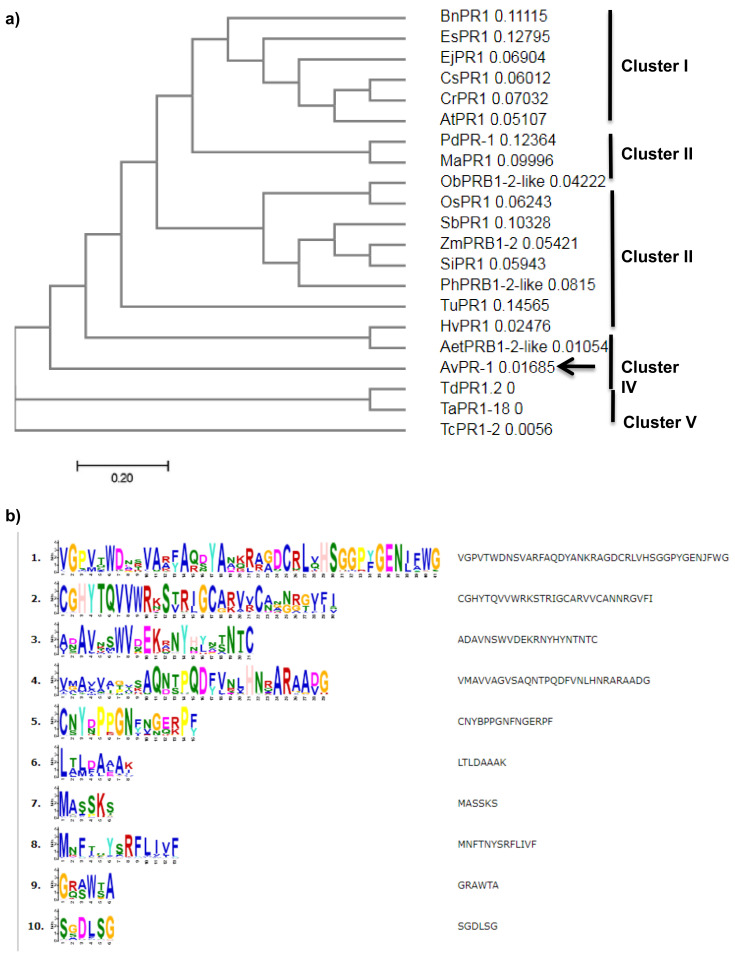
(**a**) Phylogenetic analysis of 21 different pathogen-related proteins oat (*GenBank OP132412*), *Triticum turgidum* subsp. durum (MK570869.1), *Triticum aestivum* (XP_044433901.1), *Aegilops tauschii* subsp. tauschii (XP_020170282.1), *Triticum urartu* (EMS45472.1), *Phoenix dactylifera* (XP_008796972.2), *Hordeum vulgare* subsp. vulgare (BAK01044.1), *Triticum dicoccoides* (XP_037463373.1), *Panicum hallii* (XP_025795067.1), *Oryza brachyantha* (XP_006661674.1), *Zea mays* (XP_008657154.1), *Eutrema japonicum* (BAF03626.1), *Camelina sativa* (XP_010467245.1), *Arabidopsis thaliana* (NP_179068.1), *Eutrema salsugineum* (XP_006409652.1), *Capsella rubella* (XP_006299028.1), *Brassica napus* (XP_013733404.1), *Sorghum bicolor* (XP_002465112.1), *Setaria italica* (XP_004983398.1), *Musa acuminata* (ABK41053.2), *Oryza sativa* Japonica Group (XP_015613013.1) using the neighbor-joining (NJ) tree for the PR1 query, generated using the Cluster Omega program. (**b**) LOGO presentation of conserved segments in PR-1 proteins in plants. Amino acids are grouped by color according to their physiochemical properties. The height of the amino acids corresponds to their conservation at that position.

**Figure 4 plants-11-02284-f004:**
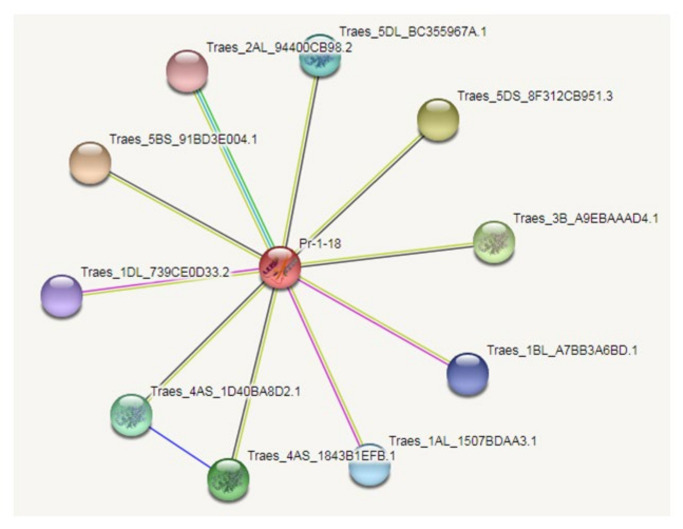
Protein–protein interaction network of AvPR-1 protein.

**Figure 5 plants-11-02284-f005:**
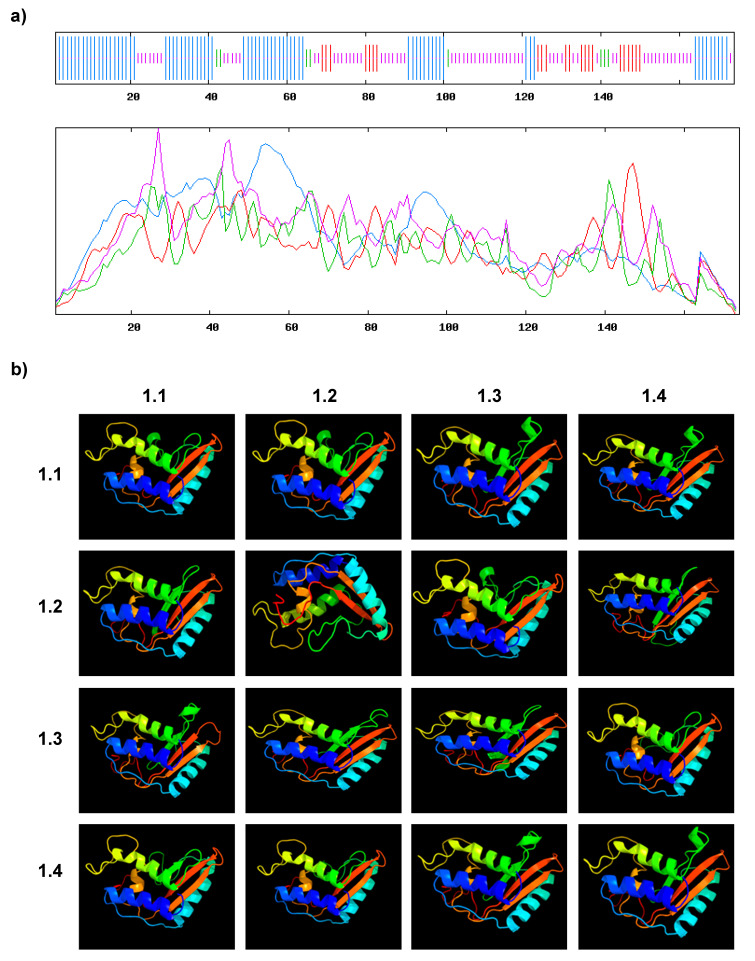
(**a**) 2D structure presentation of AvPR-1 as revealed by the SOPMA server. The alpha helixes are presented in blue, the extended strands are presented in red, the beta turns are presented in green, and the random coils are presented in yellow. (**b**) 3D structure of different PR-1 proteins used in this work as revealed by the PHYRE2 database (1.1 AvPR-1; 1.2 HvPR-1; 1.3 TdPR1.2; 1.4 TaPR1; TcPR1.2; AetPR1; PhPR1-like, TuPR1; EjPR-1; ObPRB1-2-like; SbPR1; SbPR1; ZmPRB1-2; SiPR1; MaPR1; EsPR1; CsPR1; AtPR1; OsPR1; PdPR1).

**Figure 6 plants-11-02284-f006:**
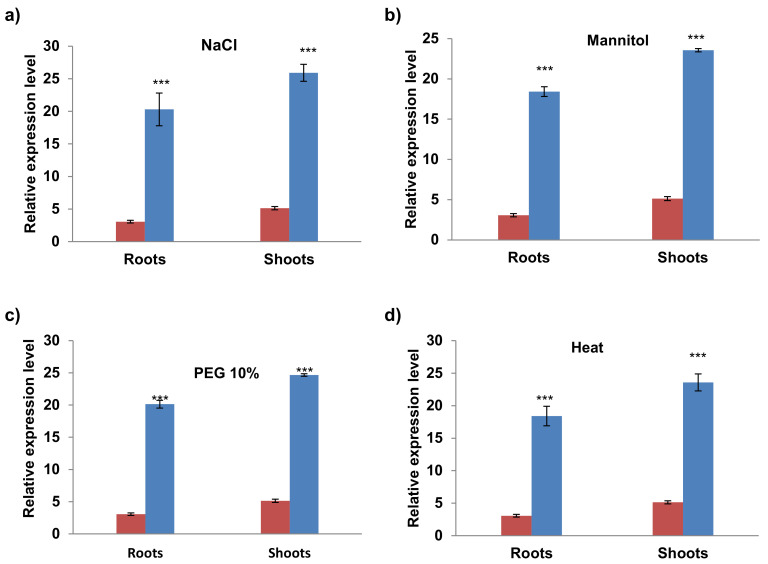
qRT-PCR expression analysis of AvPR-1 gene under different abiotic stresses (**a**) *salt,* (**b**) *mannitol,* (**c**) PEG 10%, and (**d**) heat. The red bars represent the expression level of the AvPR-1 gene under standard conditions, and the blue bars represent the expression level of the AvPR-1 gene under stressed conditions. (***) indicates value significantly different from the control. Statistical significance was assessed by applying the student *t*-test at *p* < 0.01.

**Figure 7 plants-11-02284-f007:**
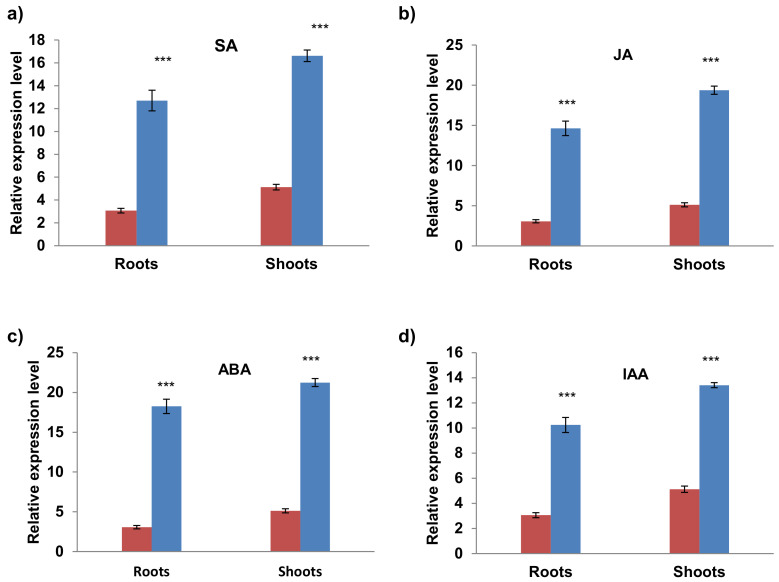
qRT-PCR expression analysis of AvPR-1 gene under different hormonal stresses (**a**) *Salycilic acid (SA)* (**b**) *Jasmonic acid (JA) (***c**) abscisic acid (ABA), and (**d**) IAA. The red bars represent the expression level of the AvPR-1 gene under standard conditions, and the blue bars represent the expression level of the AvPR-1 gene under stressed conditions. (***) indicates value significantly different from the control. Statistical significance was assessed by applying the student *t*-test at *p* < 0.01.

**Table 1 plants-11-02284-t001:** Comparison between different isolated PR-1 from plants using ProtParam tool (http://web.expasy.org/protparam/; accessed on 1 May 2022).

Protein	MW	Number of aa	Number of Negatively Charged Residues	Number of Positively Charged Residues	Grand Average of Hydropathicity (GRAVY)	Aliphatic Index	PI
AvPR1	18.89	174	12	18	−0.288	63.45	9.19
TdPR1	18,836.12	174	12	17	−0.238	65.11	9.02
TcPR1.2	18,850.15	174	12	17	−0.237	65.11	9.02
AetPR1	18,658.97	172	12	18	−0.273	66.40	9.17
HvPR1	18,969.98	172	12	19	−0.333	63.02	9.32
PhPR1-like	18,898.31	174	15	20	−0.190	70.80	9.00
TuPR1	18,330.67	167	10	21	−0.362	67.19	9.86
EjPR-1	17,668.02	161	10	16	−0.268	81.74	9.1
ObPRB1-2-like	18,458.88	172	11	20	−0.182	65.99	9.51
SbPR1	19,094.56	179	14	20	−0.152	66.15	9.10
BnPR-1	17,771.98	162	10	16	−0.315	78.27	9.02
ZmPRB1-2	19,156.67	179	13	21	−0.189	68.38	9.50
SiPR1	19,045.54	176	12	21	−0.164	71.14	9.0
MaPR1	17,308.30	162	8	11	−0.204	66.36	8.49
EsPR1	17,706.92	161	15	16	−0.322	81.74	7.58
PdPR1	17,470.43	162	11	11	−0.182	66.36	6.93
CsPR1	17,697.91	161	8	14	−0.268	73.85	9.08
AtPR1	17,676.94	161	10	16	−0.288	73.85	9.08
OsPR1	18,743.06	176	12	18	−0.227	66.65	9.10

**Table 2 plants-11-02284-t002:** Secondary structure analysis of AvPR-1 and other plant PR-1 using SOPMA program.

Protein	α-Helices	Extended Strands	Random Coils	β-Turns
AvPR-1	55	8	81	30
TdPR1.2	55	8	81	30
HvPR1	68	8	70	26
PhPR1	66	8	69	31
ObPR1	64	7	75	26
SbPR1	64	7	81	27
DoPR1	47	8	72	31
PvPR1	66	10	72	28
ZmPR1	68	10	71	31
SiPR1	66	8	71	31
MaPR1	55	7	70	30
AsPR1	61	8	71	26
PdPR1	57	9	69	27
CsPR1	55	9	68	31
TaPR1	55	8	81	30
AtsPR1	68	8	71	25
OsPR1	60	8	82	26
ClPR1	58	6	72	28

## Data Availability

The data generated and analyzed during this study are included in this article.

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
