# Peer review of "Isolation and Characterization of a Novel Pathogenesis-Related Protein-1 Gene (AvPR-1) with Induced Expression in Oat (Avena sativa L.) during Abiotic and Hormonal Stresses"

_plants, 2022, doi:10.3390/plants11172284_

Round 1

Reviewer 1 Report

Report 1   Reviewer’s Information (will not be revealed to authors)   Please note that you are in the reviewer board of Prosthesis. If you would like to update your profile, please contact Prosthesis Editorial Office to update your profile. ORCID  0000-0002-6157-2462 [What is this?] Name Dr. Shrikant Pawar Email [email protected] Website http://sites.gsu.edu/spawar2/ Affiliation Department of Genetics, Yale University, New Haven, CT 06511, USA Research Keywords bioinformatics; computational biology; data science; machine learning   * Open Review  I would not like to sign my review report
 I would like to sign my review report   This journal operates an open review policy, so your review report may be published. By signing your report your name will appear alongside the published report. Note that the authors will not know your identity until the paper is published.   Recommendations for Authors (will be shown to authors) The following questions do not substitute for specific comments made for authors. Please give further details in the comments for authors box below.      
  Yes Can be improved Must be improved Not applicable
Does the introduction provide sufficient background and include all relevant references?
Are all the cited references relevant to the research?
Is the research design appropriate?
Are the methods adequately described?
Are the results clearly presented?
Are the conclusions supported by the results?

* English language and style  Extensive editing of English language and style required

 Moderate English changes required

 English language and style are fine/minor spell check required

 I don't feel qualified to judge about the English language and style

  Recommendations for Editors (will not be shown to authors) If you answered yes to any of the following questions, please give details in the comments for editors box below.      
  Yes No
Do you have any potential conflict of interest with regards to this paper?
Did you detect plagiarism?
Did you detect inappropriate self-citations by authors?
Do you have any other ethical concerns about this study?

Ratings High Average Low No Answer
* Originality / Novelty
* Significance of Content
* Quality of Presentation
* Scientific Soundness
* Interest to the readers
* Overall Merit

Author Response

Reviewer 1:

  • Does the introduction provide sufficient background and include all relevant references? à must be approved
  • Are the methods adequately described ?--< Can be approved
  • Are the results clearly presented?--> can be approved
  • Are the conclusions supported by the results?--> must be approved

We have improved the manuscript as suggested by the reviewer 1.

Reviewer 2 Report

The authors of this article isolated  PR-1 protein from Oat plants and characterized using insilco tools and its expression analysis by qPCR.

Majorly, the manuscript has many typographical errors and language issues. it has to be thoroughly revised. I have stated many such errors in the MS.

Authors may consider expressing the oat PR-1 gene in arabidopsis and to evaluate further functionality to various stress conditions.           

Author Response

Reviewer 2:

  • Does the introduction provide sufficient background and include all relevant references? à must be approved
  • Are the methods adequately described ?--< Can be approved
  • Are the results clearly presented ?--> can be approved
  • Are the conclusions supported by the results?--> must be approved

We have improved the manuscript as suggested by the reviewer 2.

  • The authors of this article isolated PR-1 protein from Oat plants and characterized using in silco tools and its expression analysis by qPCR. Majorly, the manuscript has many typographical errors and language issues. It has to be thoroughly revised. I have stated many such errors in the MS.

All typographical errors and language issues are checked in the manuscript.

  • Authors may consider expressing the oat PR-1gene in Arabidopsis and to evaluate further functionality to various stress conditions.   

We thank the reviewer 2 for his comment. Effectively, transgenic plants over-expressing AvPR-1 gene are actually generated and their characterization are underway and will be the subject of a second paper.

Reviewer 3 Report

In the manuscript entitled “Isolation and characterization of a novel Pathogenesis Related Protein-1 gene (AvPR-1) with induced expression in Oat (Avena sativa L.) during abiotic and hormonal stresses”, the authors have cloned the pathogenesis-related proteins 1 gene (AvPR-1) and characterized it bioinformatically. qRT-PCR indicated that AvPR-1 responds to many abiotic and hormonal stresses. Because PR-1 proteins are very important in plant responses to biotic and abiotic stresses. This gene may be used to regulate oat responses to biotic and abiotic stresses. The paper is well written. The conclusions were consistent with the evidence.

The shortcoming was that no fundamental finds were presented. The interaction network of AvPR-1 protein was not validated experimentally. 

Minors:

(1) Line 209: please add a distance scale bar or indicate the distance in Figure 3 (a).

(2) Line 291: Figure 6 is not clear. Please indicate the meaning of the blue bar and the red bar. The difference should be determined statistically.

(3) Line 307: Figure 7 is not clear. Please indicate the meaning of the blue bar and the red bar again. The difference should also be determined statistically.

(4) The authors should describe the qRT-PCR method, including the internal control, primer sequence, and so on.

Author Response

Reviewer 3:

  • Are the methods adequately described? à must be approved
  • Are the results clearly presented?--> can be approved

We tried to improve the manuscript as suggested by the reviewer 3.

  • In the manuscript entitled “Isolation and characterization of a novel Pathogenesis Related Protein-1 gene (AvPR-1) with induced expression in Oat (Avena sativa L.) during abiotic and hormonal stresses”, the authors have cloned the pathogenesis-related proteins 1 gene (AvPR-1) and characterized it bioinformatically. qRT-PCR indicated that AvPR-1 responds to many abiotic and hormonal stresses. Because PR-1 proteins are very important in plant responses to biotic and abiotic stresses. This gene may be used to regulate oat responses to biotic and abiotic stresses. The paper is well written. The conclusions were consistent with the evidence.
  • The shortcoming was that no fundamental finds were presented. The interaction network of AvPR-1 protein was not validated experimentally. 

We would thank the reviewer 3 for these comments. This is the first PR gene (AvPR1) isolated from Oat (Avena sativa L.). Moroever, the genome of Avena is not yet determined and little data are valuable online. More experiments are needed to understand the role played by AvPR-1 to respond to abiotic and biotic stress.

Minors:

  • Line 209: please add a distance scale bar or indicate the distance in Figure 3 (a).

A distance scale was added to the figure 3a

  • Line 291: Figure 6 is not clear. Please indicate the meaning of the blue bar and the red bar. The difference should be determined statistically.

The colors of the histograms are indicated in the legends and statistical analyses are added in the figure 6.

  • Line 307: Figure 7 is not clear. Please indicate the meaning of the blue bar and the red bar again. The difference should also be determined statistically.

The colors of the histograms are indicated in the legends and statistical analyses are added in the figure 7.

 (4) The authors should describe the qRT-PCR method, including the internal control, primer sequence, and so on.

We totally agree with the reviewer and the following paratgraph was added to the text:

2.7. RNA Extraction and Quantitative Real-Time Reverse Transcription PCR (qRT-PCR) :

‘’Total RNA was extracted from individual roots, and leaves (0.5 g of each tissue) …………………………………………………………………………………Relative expression ratios from three independent experiments (three biological repetitions) are reported.’’
